# Efficient biosynthesis of nucleoside cytokinin angustmycin A containing an unusual sugar system

Le Yu[1,4], Wenting Zhou[1,4], Yixuan She[1,4], Hongmin Ma[1], You-Sheng Cai[1], Ming Jiang [2], Zixin Deng [1,2], Neil P. J. Price [3✉] & Wenqing Chen [1✉]

Angustmycin A has anti-mycobacterial and cytokinin activities, and contains an intriguing structure in which an unusual sugar with C5′-C6′ dehydration is linked to adenine via an *N*-glycosidic bond. However, the logic underlying the biosynthesis of this molecule has long remained obscure. Here, we address angustmycin A biosynthesis by the full deciphering of its pathway. We demonstrate that AgmD, C, A, E, and B function as D-allulose 6-phosphate 3-epimerase, D-allulose 6-phosphate pyrophosphokinase, adenine phosphoallulosyltransferase, phosphoribohydrolase, and phosphatase, respectively, and that these collaboratively catalyze the relay reactions to biosynthesize angustmycin C. Additionally, we provide evidence that AgmF is a noncanonical dehydratase for the final step to angustmycin A via a self-sufficient strategy for cofactor recycling. Finally, we have reconstituted the entire six-enzyme pathway in vitro and in *E. coli* leading to angustmycin A production. These results expand the enzymatic repertoire regarding natural product biosynthesis, and also open the way for rational and rapid discovery of other angustmycin related antibiotics.

[1] Key Laboratory of Combinatorial Biosynthesis and Drug Discovery, Ministry of Education, and School of Pharmaceutical Sciences, Wuhan University, 430071 Wuhan, China. [2] State Key Laboratory of Microbial Metabolism, and School of Life Sciences and Biotechnology, Shanghai Jiao Tong University, 200240 Shanghai, China. [3] US Department of Agriculture, Agricultural Research Service, National Center for Agricultural Utilization Research, Peoria, IL, USA. [4]These authors contributed equally: Le Yu, Wenting Zhou, Yixuan She. ✉email: neil.price@ars.usda.gov; wqchen@whu.edu.cn

The nucleoside antibiotics are a large and structurally diverse group of compounds that often exert biological effects by mimicking naturally occurring nucleosides or nucleotides[1,2]. They are typically comprised of a modified nucleoside base, *N*-glycosidically linked to a ribosyl sugar residue, analogous to the structural characteristics of biological purine or pyrimidine ribonucleosides. The angustmycins, and related angustmycin A (**1**, also called decoyinine) and angustmycin C (**2**, also called psicofuranine), are adenosine analogs, but are unusual in having a six-carbon ketose sugar, β-D-psicofuranosyl, in place of the more usual ribofuranose motif (Fig. 1)[3,4]. The *N*-glycosidic linkage of the ketose sugar to adenine is via an unusual β-2-*N*-ketoglycosidic bond that is rarely found elsewhere in nature[5]. **1** differs from **2** in that it has an *exo*-5,6-ene bond similar to the *exo*-glycal intermediates found in the biosynthesis of tunicamycins[6]. The 5′,6′-ene group of **1** precludes any of the phosphorylation of the 6′-OH, which is possible with **2**, and potentially also effects the local planarity of the furanosyl ring.

Several chemical syntheses have been reported for the angustmycin family[7–11], including 4-thio angustmycin C[12], although considerably less is known regarding the biosynthesis. Early radiolabeling studies by Suhadolnik and co-workers have shown that the 6′-deoxy-D-erythro-2′,5′-hexodiulose sugar motif of **1** arises directly from D-[1-¹⁴C] glucose or uniformly ¹⁴C-labeled D-fructose[13,14]. Also, **2** labeled with ¹⁴C in the adenine group and at C-6 of the D-allulose is directly converted to **1** by growing cells of *Streptomyces hygroscopicus*[14].

The biological effects induced by angustmycin-related antibiotics have been studied for over 50 years[15,16]. **1** is a potent inhibitor of GMP (guanosine monophosphate) synthesis in Gram-positive bacteria and is commonly used as a biochemical tool to induce bacterial sporulation[17,18]. More recently, the inhibition of the murine GMP synthase by **1** has been shown to suppress melanoma cell invasion and tumorigenicity in immunocompromised mice, suggesting the potential of angustmycins as an anti-melanoma agent[19,20].

Angustmycin A (**1**) was also recently found to be a promising cytokinin (called Linfusu) that induces adventitious root or bud differentiation[21]. Cytokinins, as plant-specific chemical messengers (hormones), usually participate in the regulation of the cell cycle and several development processes during plant growth[22]. Adenine-derived cytokinins are abundant in nature, including zeatin with an adenine base and a five-carbon isopentenyl side

chain[23]. The naturally occurring *trans*-zeatin was initially isolated from maize endosperm in the early 1960s[24] and several other nucleoside cytokinins were identified later from plants or microbial metabolites[25,26]. Like cytokinin, **1** has been previously demonstrated to promote the growth of several plants, including *Panax notoginseng*, *Siraitia grosvenorii*, and *Triticum aestivum*[27].

In the present paper, we address the mechanism for the biosynthesis of the angustmycin family of nucleotide antibiotics. We demonstrate a six-enzyme pathway for the biosynthesis of **1**, which highlights an unusual dehydration step via a self-sufficient economical strategy for cofactor recycling. We have reconstituted the entire six-enzyme pathway both in vitro and in *Escherichia coli*, leading to **1**. The study provides an understanding of the logic underlying **1** biosynthesis, expanding our knowledge of natural product biosynthesis, and also paves the way for further engineering of the pathway for rational enhancement of **1** production.

## Results and discussion

**Identification of the angustmycin gene cluster from *S. angustmyceticus* JCM 4053.** To identify the gene cluster for angustmycin biosynthesis, we investigated the genomes of two independent angustmycin producer strains, *Streptomyces angustmyceticus* JCM 4053 and *Streptomyces decoyicus* NRRL 2666, whose capabilities of angustmycins production were confirmed by liquid chromatography-mass spectrometry (LC-MS) (Supplementary Fig. 1a–d). Since adenine is the direct precursor for the corresponding adenosyl motif of angustmycins[13], we initially used Ari9 (the adenosine phosphorylase in the aristeromycin pathway) as a query sequence to perform BlastP analysis against *S. angustmyceticus* JCM 4053 genome (accession no. CP082945, BLAG01000004). This identified a homolog with 45% identity to Ari9, but the surrounding genes (lotus tag, K7396_21205-K7396_21225; accession no. CP082945) are apparently unrelated to angustmycin biosynthesis. We deduced that angustmycin biosynthesis may use a biosynthetic strategy similar to that of *trans*-zeatin to provide the purine-related moiety[28], and we subsequently used the enzyme LOG (Lonely guy; accession no. AK071695) as the query sequence to conduct BlastP analysis (Supplementary Fig. 1e), leading to the discovery of five candidate enzymes. The surrounding regions of these candidates revealed a gene cluster (*agm*, accession no. MZ151497) that encodes

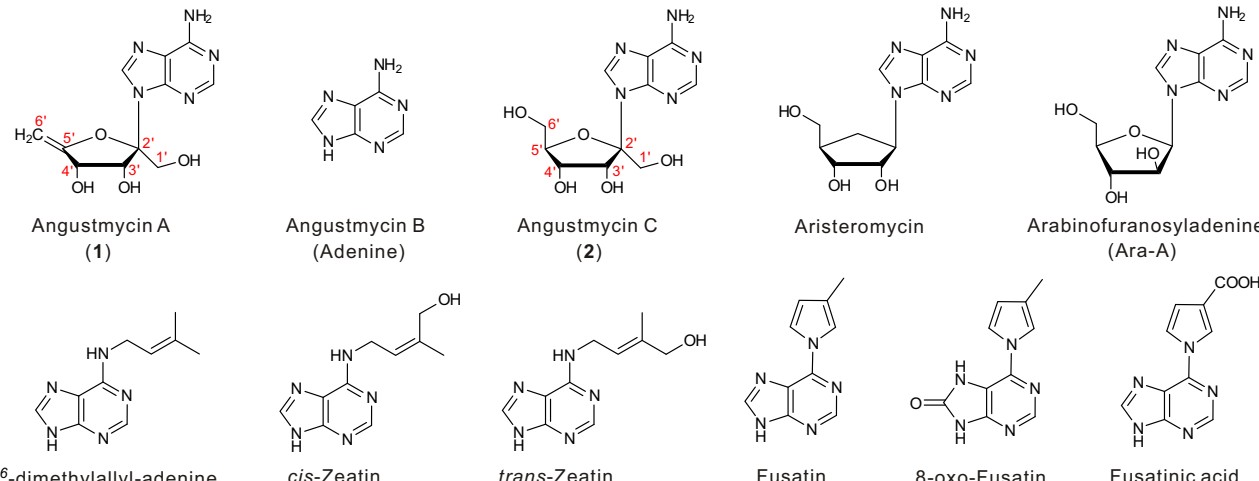

**Fig. 1 Chemical structures of angustmycin-related purine nucleoside antibiotics and cytokinins.** Relevant purine nucleosides (top) include three components of angustmycins and their analogs aristeromycin and Ara-A; purine nucleoside cytokinins (bottom) contain structural moieties similar to the purine base.

enzymes related to allulose sugar biosynthesis and assembly. Furthermore, using AgmA as the query sequence led to the identification of a homologous gene cluster (accession no. MZ151498) in *S. decoyicus* NRRL 2666 genome (accession no. CP082301) with the identical organization and significant homology. Hence, these initial findings suggested that the target gene cluster (*agm*) may be directly responsible for angustmycin biosynthesis (Fig. 2a).

To correlate the *agm* gene cluster to angustmycin biosynthesis, the gene cluster from *S. angustmyceticus* JCM 4053 was directly cloned by two-fragment PCR coupled with Gibson assembly. The subsequent gene cluster and its variant (pCHW501 and pCHW501Δ*agmF*) were individually introduced into *Streptomyces coelicolor* M1154 (Supplementary Fig. 2a, b). After confirmation, the resultant conjugants were fermented for metabolite analysis. A bioassay indicated that the metabolite samples of M1154::pCHW501 and JCM 4053 show apparent inhibition against the indicator strain *Mycobacterium smegmatis*, while the samples of M1154::pCHW501Δ*agmF* and M1154::pSET152 lacked the relevant bioactivity (Fig. 2b). High-performance liquid chromatography (HPLC) analysis indicated that the sample from *S. coelicolor* M1154::pCHW501 generated target peaks corresponding to those of **2** and **1** of JCM 4053 metabolites, while these peaks were absent from the metabolites of the negative control (*S. coelicolor* M1154::pSET152). LC-MS analysis revealed that the target peaks of M1154::pCHW501 gave rise to distinctive $[M + H]^+$ ions at $m/z$ 298.1144 (with major fragment ions at $m/z$ 280.0587 and 136.0617) and $m/z$ 280.1040 (with fragment ions at $m/z$ 262.0929 and 136.0618), which are fully consistent with the expected fragmentation patterns of **2** and **1** (Supplementary Fig. 3b, c). More interestingly, the metabolites recovered from M1154::pCHW501Δ*agmF* selectively produced the **2** peak, but not **1**, suggesting that the *agmF* gene is likely responsible for the conversion of **2** to **1** (Fig. 2c and Supplementary Fig. 3a).

To establish the identities of the target metabolites accumulated by M1154::pCHW501 and M1154::pCHW501Δ*agmF*, they were HPLC purified for 1D and 2D nuclear magnetic resonance (NMR) analysis. The proton-NMR ($^1$H-NMR) and heteronuclear multiple quantum coherence NMR spectra of the metabolite from M1154::pCHW501Δ*agmF* displayed two methylenes, five methines (including two sp$^2$ carbons), and four quaternary carbons (including three sp$^3$). Detailed analysis of the 1D and 2D NMR data led to the identification of this metabolite as **2**, which was further confirmed by $^1$H-$^1$H COrrelated SpectroscopY correlations of H-3′–H-4′–H-5′–H-6′, and by heteronuclear multiple bond correlations (HMBC) of H-1′ with C-2′ and C-3′; H-8 with C-4 and C-5; and H-2 with C-4, C-5, and C-6 (Supplementary Fig. 4, 5 and Supplementary Table 1). Comparison of the 1D and 2D NMR data of the M1154::pCHW501 metabolites with those of **2** suggested structural similarities, except that the hydroxymethyl group at C-5′ in **2** was replaced by an exocyclic double bond. This was further confirmed by HMBC

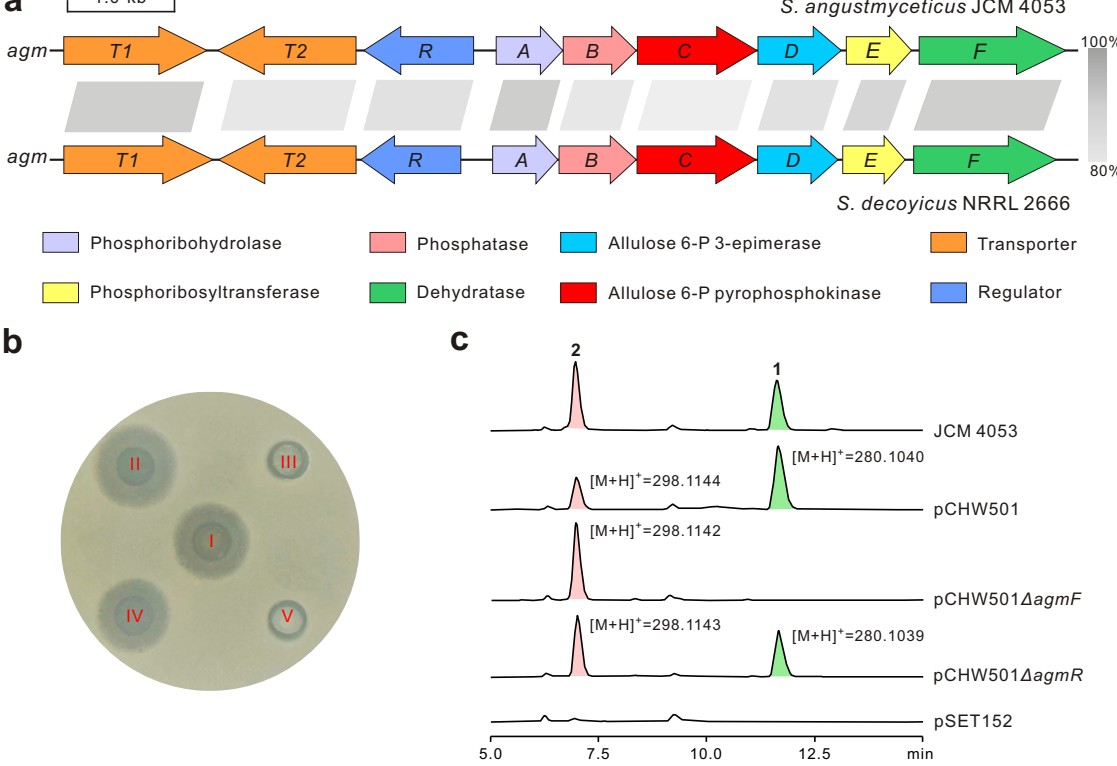

**Fig. 2 Genetic organization and verification of the angustmycin biosynthetic gene cluster. a** Genetic organization of the angustmycin biosynthetic gene cluster. The *agm* gene cluster is responsible for angustmycin biosynthesis in *S. angustmyceticus* JCM 4053 and *S. decoyicus* NRRL 2666. The two gene clusters in different strains harbor significant homologies, and the gray shaded bars represent the identities of the two corresponding enzymes encoded by the two gene clusters. **b** Bioassays of the metabolites produced by related recombinants of *S. coelicolor* M1154. The indicator strain is *Mycobacterium smegmatis* mc$^2$155. The Roman numerals I, II, III, IV, and V represent JCM 4053, pCHW501, pCHW501Δ*agmF*, pCHW501Δ*agmR*, and pSET152, respectively, and the denotations of them correspond to those shown in (**c**). **c** HPLC analysis ($λ = 254$ nm) of the metabolites produced by related recombinants of *S. coelicolor* M1154. JCM 4053, **1** and **2** produced by *S. angustmyceticus* JCM 4053 as positive controls ; pCHW501, the metabolites of the recombinant *S. coelicolor* M1154 containing pCHW501; pCHW501Δ*agmF*, the metabolites of the recombinant *S. coelicolor* M1154 containing pCHW501Δ*agmF*; pCHW501Δ*agmR*, the metabolites of the recombinant *S. coelicolor* M1154 containing pCHW501Δ*agmR*; pSET152, the metabolites of the recombinant *S. coelicolor* M1154 containing pSET152 as a negative control. Allulose 6-P, Allulose 6-phosphate.

correlations from H-6′ to C-4′ and C-5′ (Supplementary Figs. 6 and 7 and Supplementary Table 2). Accordingly, the structures of the target metabolites were identical to those of **2** and **1** as shown in Fig. 1. Taken together, these data also demonstrated that the target gene cluster is responsible for the biosynthesis of angustmycin.

**The nine-gene cluster *agm* is required for angustmycin biosynthesis.** In silico analysis revealed a target 9.8 kb region (pCHW501) containing nine genes that are deduced to be involved in angustmycin biosynthesis. Of these, *agmA-F*, constituting the structural genes, are important for angustmycin biosynthesis on the basis of genetic and in silico analysis, and their functions are described in Supplementary Table 3. Moreover, AgmT1 and AgmT2 are annotated as MFS (major facilitator superfamily) transporters. In general, MFS proteins facilitate the transport of a variety of substrates across cytoplasmic or internal membranes[29], and therefore, AgmT1 and AgmT2 are proposed to play the functional role of exporting angustmycins out of cells. AgmR shows 50% identity to A6A06_38595 (LacI family regulator) of *Streptomyces* sp. CB02923, and may have a regulatory role by binding to the promoter region during angustmycin biosynthesis. To establish the role of *agmR*, the pCHW501Δ*agmR* variant was conjugated into *S. coelicolor* M1154, and the resultant recombinant was then fermented for metabolite analysis (Supplementary Fig. 2c and Supplementary Table 4). HPLC analysis indicated that the sample from M1154::pCHW501Δ*agmR* was capable of producing two peaks corresponding to **2** (major component) and **1**, which are quantitatively different from that of M1154::pCHW501, implying that there may be an endogenous regulator in *S. coelicolor* M1154, which also participates in upregulating the transcriptional level of the biosynthetic genes to control the ratio of **2** and **1** (Fig. 2c and Supplementary Fig. 3a, d). Related studies to investigate the regulatory mechanism during angustmycin biosynthesis are now underway in our laboratory.

**AgmA functions as an AMP phosphoribohydrolase.** As indicated by in silico analysis, AgmA shows considerable homology (54% identity) to the LOG gene of *Oryza sativa Japonica* Group (japonicum cultivar rice), which is a well-characterized phosphoribohydrolase involved in *trans*-zeatin biosynthesis, so we proposed that AgmA performs a similar function in angustmycin biosynthesis (Supplementary Fig. 8a). To determine if AgmA indeed fulfills this role, it was overexpressed and purified to near homogeneity from *E. coli* BL21(DE3). The in vitro activity was tested using AMP/dAMP as substrate. Consistent with our expectation, the HPLC results indicated that AgmA is capable of consuming AMP/dAMP as a substrate to produce a characteristic peak for adenine, which was absent from the negative control (Fig. 3a and Supplementary Fig. 8b–f). Further LC-MS analysis showed that the target peak gave rise to a distinctive $[M + H]^+$ ion for adenine at $m/z$ 136.0613 (with major fragment ions at $m/z$ 118.8611 and 93.8582) and $m/z$ 136.0613 (with major fragment ions at $m/z$ 118.8811 and 93.9553), fully corresponding with those of the adenine authentic standard (Supplementary Fig. 9a–c). We then determined the specificity of AgmA against various other substrates related to AMP, including ATP, ADP, dATP, dADP, adenosine, and deoxyadenosine. The HPLC results from these experiments indicated that the AgmA enzyme could also recognize ATP, ADP, and dADP as substrates, but with lower relative activity (Supplementary Fig. 9d). These data establish that AgmA functions as an AMP phosphoribohydrolase for the supply of adenine.

The LOG family of cytokinin-activating enzymes (InterPro: IPR005269), as revealed by BlastP analysis, is widespread among

the three kingdoms of life (with ca. 9000 sequences retrieved from the UniProt database). These sequences were incorporated using an enzyme similarity tool (EFI-EST) to generate a sequence similarity network (SSN)[30] with the UniRef90 database based on an alignment score of 80 (Fig. 3b). These results indicated that AgmA homologs are more widely distributed in bacterial genomes, and also in plant genomes. The related enzymes of actinobacterial origin, including the AgmA described in this study, cluster in the network (highlighted in red). These cytokinin-activating family proteins commonly contain a conserved PGGxGTxxE motif, which differs from the canonical lysine decarboxylases[31]. Their proposed catalytic activity in plants is to convert inactive cytokinin nucleotides to active forms. Foreseeably, the AgmA-related enzyme may be of considerable value for plant cytokinin studies and applications.

**Angustmycin A biosynthesis features an unusual final dehydration step.** The C-5′–C-6′ double bond present in the sugar moiety of **1** is chemically intriguing. In our initial studies, the truncated *agm* gene cluster (pCHW501Δ*agmF*) was revealed to confer the host cell *S. coelicolor* M1154 with the capability of selectively producing **2**, implying that the candidate enzyme AgmF is likely to govern the final dehydration step of **2** to **1**. Surprisingly, the further bioinformatic analysis showed that AgmF was annotated as an *S*-adenosylhomocysteine (SAH) hydrolase, which usually catalyzes the reversible hydrolysis of SAH to adenosine and homocysteine[32,33]. SAH hydrolase is a ubiquitous enzyme that plays a central role in cellular methylation processes by maintaining the intracellular balance between SAH and *S*-adenosylmethionine[34]. To determine if AgmF also fulfills the role of a SAH hydrolase, it was overexpressed and purified from *E. coli* BL21(DE3), and we subsequently evaluated the enzymatic assay of AgmF using SAH as substrate. More surprisingly, the results showed that the AgmF enzyme could not recognize it as the substrate. We, therefore, reinvestigated the enzymatic role of AgmF, and inferred that **2** is more likely to be its real substrate. Accordingly, we tested its activity with **2** in the presence or absence of the exogenous NAD$^+$ (nicotinamide adenine dinucleotide) cofactor, and the HPLC results indicated that **2** is indeed dehydrated to form **1** (Supplementary Fig. 10a). The identity was further confirmed by LC-MS, as it could give rise to $[M + H]^+$ ions at $m/z$ 280.1032 (with main fragment ions at $m/z$ 262.0833 and 136.0000) and $m/z$ 280.1031 (with main fragment ions at $m/z$ 262.0000 and 135.9167), which fully matched to those of a **1** authentic standard. More interestingly, we also report that AgmF is able to maintain the activity without the addition of exogenously NAD$^+$ cofactor, implying that AgmF is a self-sufficient enzyme in which the NAD$^+$ cofactor is tightly bound within the activity pocket (Fig. 4a and Supplementary Fig. 12a). To further test this hypothesis, the AgmF enzyme was heat-treated to release the potential cofactor, which was then identified by HPLC as a distinctive NAD$^+$ peak, whose identity was further determined by LC-MS (Supplementary Fig. 10b–f). To compare the difference in the enzymatic activity with or without NAD$^+$, we conducted a time-course experiment, illustrating that the AgmF reaction with NAD$^+$ occurs at a higher rate in the early reaction stage (Supplementary Fig. 10g). Furthermore, we noticed that substrate **2** was not be used up in the AgmF reaction, implying that this enzyme may catalyze a reversible reaction. Hence, AgmF may catalyze a reversible reaction from **1** to **2** with the chemical equilibrium shifted to **1** formation (Fig. 4b and Supplementary Fig. 12b).

We next determined the kinetic parameters of AgmF for the substrate **2**, with $K_M = 1.382 \pm 0.121$ mM, and $k_{cat} = 1.902 \pm 0.061$ s$^{-1}$ (Supplementary Fig. 13a). The $K_M$ value is larger than

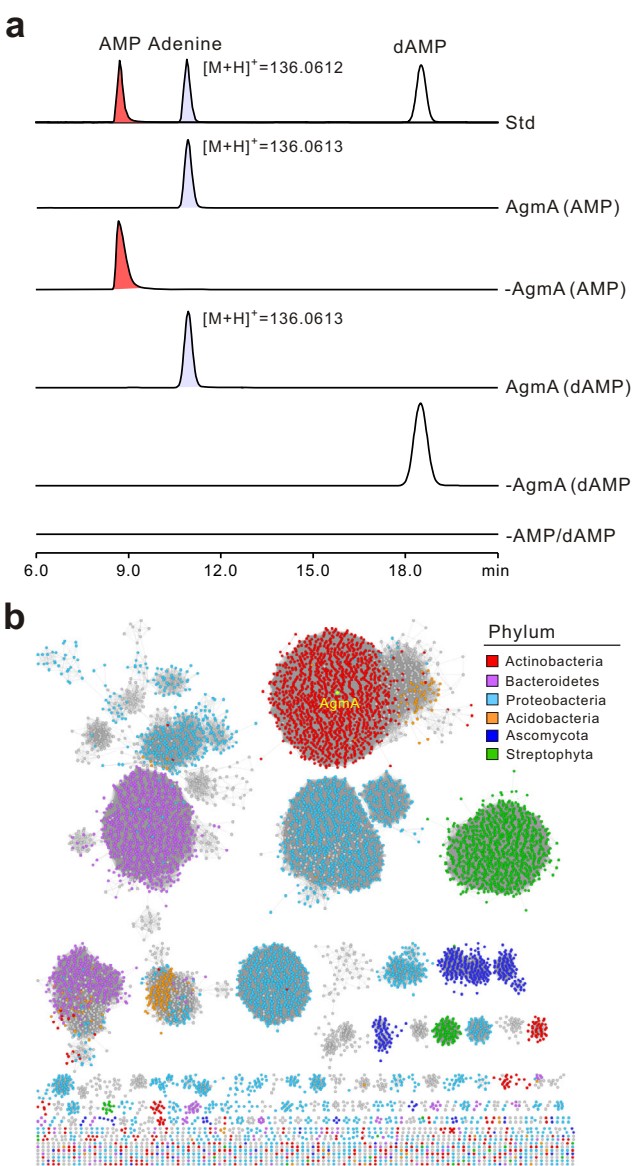

**Fig. 3 Functional characterization of AgmA as AMP phosphoribohydrolase. a** HPLC traces ($\lambda = 254$ nm) of the AgmA reactions using AMP or dAMP as the substrate. Std, the authentic standards of AMP, dAMP, and adenine; AgmA(AMP), AgmA reaction with AMP as substrate; -AgmA(AMP), reaction using AMP as substrate but without AgmA added; AgmA(dAMP), AgmA reaction with dAMP as substrate; -AgmA(dAMP), reaction using dAMP as substrate but without AgmA added; -AMP/dAMP, AgmA reaction without AMP or dAMP added. **b** Sequence similarity network (SSN) of 9058 AgmA homologs generated by Cytoscape (v3.8.2). Each node in the network represents a protein sequence and the alignment score is 70. AgmA falls into the actinobacteria cluster and is highlighted by a yellow point.

those of the reported SAH hydrolases with SAH as substrate[35]. We also evaluated the substrate specificity of AgmF against several **2** analogs, including adenosine, 2′-deoxyadenosine, 2′-amino-2′-deoxyadenosine, aristeromycin, formycin, and tubercidin, with results showing that the enzyme specifically recognizes **2** as the primary substrate (Supplementary Fig. 12d, e). To further investigate the catalytic mechanism of AgmF, a homologous structural model for AgmF was constructed based on a SAH hydrolase (containing $NAD^+$ and adenosine binding pockets) of *Mycobacterium tuberculosis* H37Rv (PDB: 3CE6, 34% identity to

AgmF)[34]. Accordingly, the side chains of Thr155, Thr156, Asn189, Thr240, Glu241, and Thr274 are identified as the candidate $NAD^+$ binding sites that are generally required for this family of SAH hydrolase-related enzymes to execute the catalytic function. Specific residues from the C terminus of the adjacent subunit also interact with the $NAD^+$, forming a covering over the binding site. This structure might partly explain the difficulty in the release of $NAD^+$ (Fig. 4c). The **2** binding to the active sites also seems to be similar to those of the model structure complexed with adenosine. His56 and His299 may form hydrogen bonds with the 6′-hydroxyl of **2**, and residues Asp156, Asp188, and Lys184 probably interacted with the hydroxyls of the furanosyl sugar (Fig. 4c and Supplementary Fig. 11). To verify these deductions, we performed selective mutational assays and determined that the AgmF variants H56A, D129A, K184A, D188A, and H299A completely abolished the activity for catalyzing **2** to **1**, thereby demonstrating that these residues are functionally essential for AgmF activity (Fig. 4d and Supplementary Fig. 12c). Hence, these combined data demonstrate that AgmF functions as an unusual $NAD^+$-dependent dehydratase. On account of the structural similarity between **2** and adenosine, we also assessed if adenosine could act as a competitive inhibitor, and the results indicated that the AgmF was found to be almost inactivated in the presence of a 3-fold excess of exogenously added adenosine (Supplementary Fig. 14).

AgmF differs from the classical family of SAH hydrolases in catalytic function, particularly in the recognition of a distinct substrate, while the key catalytic residues are conserved, suggesting that AgmF may adopt a similar catalytic mechanism to that of the typical SAH hydrolase. Concerning the dehydration reaction, several enzymatic strategies are derived from the common dehydratase domain in polyketide synthase[36], the heme iron-dependent aldoxime dehydratase[37], the dihydroxyacid dehydratase (containing a 2Fe–2S cluster)[38], and the glutamyl-tRNA-dependent dehydratase in RiPPs (ribosomally synthesized and post-translationally modified peptides) biosynthesis[39]. The generic mycobacterial SAH hydrolase has been shown to catalyze a reversible reaction via an oxidation–reduction mechanism, which is associated with a cryptic dehydration step. However, the self-sufficient strategy exploited for the $NAD^+$ cofactor recycling in the AgmF reaction is interesting. We, therefore, propose that Lys185 is likely to act as the basic residue extracting the proton from the 4′-OH group during oxidation to 4′-keto angustmycin C (**2a**) and that Lys185 is more basic because of its proximity to Asp188 and Ser190 (Fig. 4e). The carboxyl group of Asp129 can make an interaction with H-4′ and catalyze a proton abstraction at C-4′ to form **2b** (Fig. 4e). Moreover, the two histidine residues His56 and His299 are speculated to form hydrogen bonds with O6′ of **2** and may be involved in the removal of the 6′-OH of **2b** to form **2c**. Finally, the end product **1** is produced with recycling of the $NAD^+$ cofactor (Fig. 4e and Supplementary Fig. 13b, c). The leaving group proposed for the AgmF-catalyzed reaction is water, which is relatively poor when compared to the 5′-homocysteine motif for the SAH hydrolase family enzymes[32]. However, this may be promoted by more favorable conditions within the active site of AgmF or possibly involve a transient leaving group at the 5′ position (Fig. 4e and Supplementary Fig. 13c).

We also note that AgmF homologs are actually far more widely distributed in the microbial genomes than was previously supposed, which could be shown in a colored SSN analysis (Supplementary Fig. 15). A phylogenetic analysis of AgmF against other SAH hydrolases showed that the AgmF-related dehydratases formed a separate cluster from other enzymes (Supplementary Fig. 16), suggesting that AgmF represents a family of dehydratases using an unusual self-sufficient strategy for the cofactor $NAD^+$ recycling.

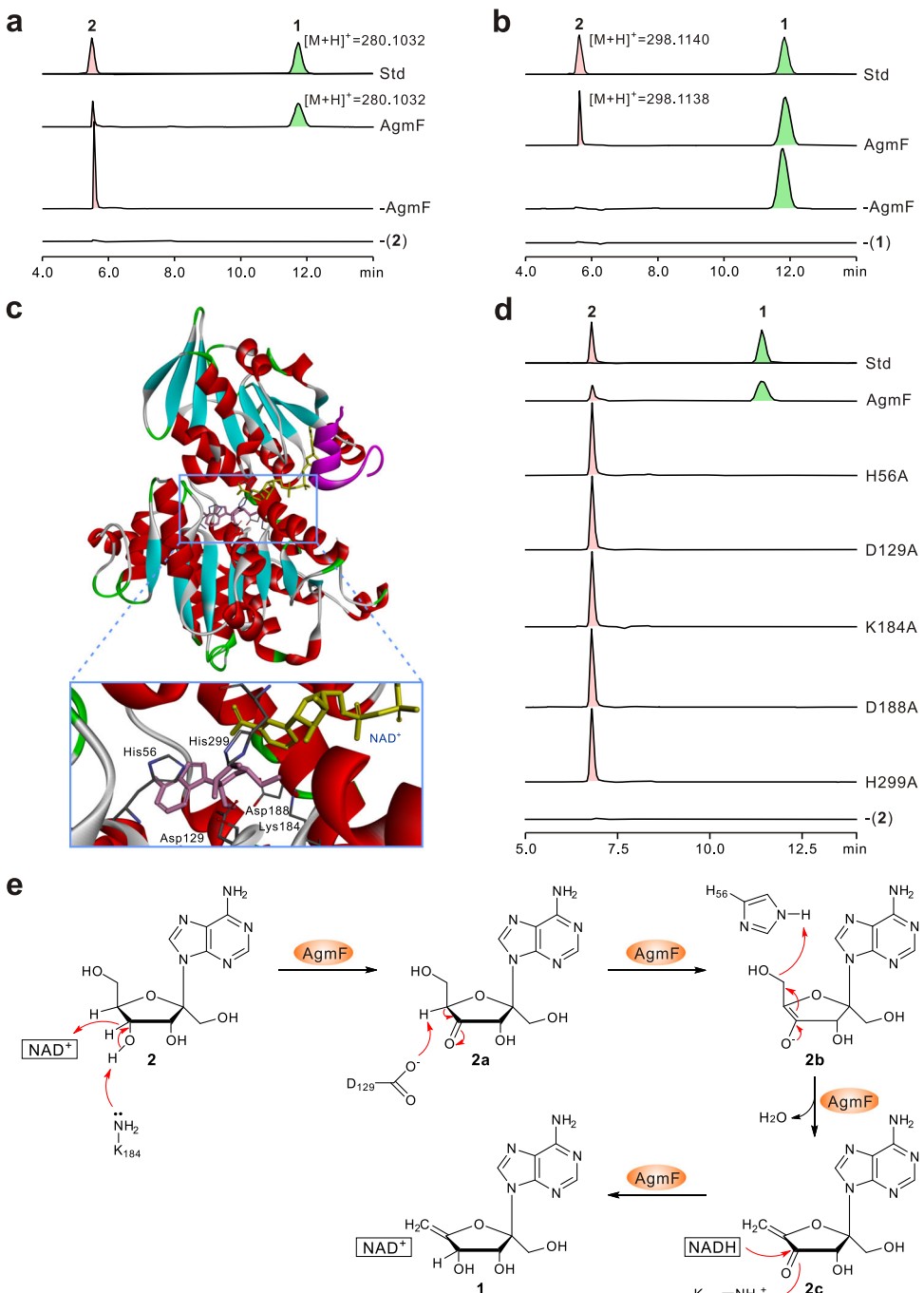

**Fig. 4 Functional characterization of AgmF as an unusual dehydratase catalyzing a reversible reaction. a** HPLC analysis ($\lambda = 254$ nm) of AgmF reaction with **2** as substrate. Std, the authentic standards of **2** and **1**; AgmF, AgmF reaction with **2** as substrate; -AgmF, reaction using **2** as substrate but without AgmF added; -(**2**), AgmF reaction without **2** added. **b** HPLC analysis ($\lambda = 254$ nm) of AgmF-catalyzed reversible reaction with **1** as substrate. Std, the authentic standards of **2** and **1**; AgmF, AgmF reaction with **1** as substrate; -AgmF, reaction using **1** as substrate but without AgmF added; -(**1**), AgmF reaction without **1** added. **c** Homology structure model of AgmF. This structure is constructed on the basis of Rv3248c from *Mycobacterium tuberculosis* H37Rv (PDB 3CE6), and the active **2** and NAD$^+$ binding sites are indicated in the rectangular region. The sites marked out are proposed to be essential for the binding of **2**. **d** HPLC analysis ($\lambda = 254$ nm) of the reactions of AgmF and its variants. Std, the authentic standards of **2** and **1**. AgmF, reaction with the intact AgmF added; lines labeled H56A, D129A, K184A, D188A, and H299A show the individual reactions of AgmF variants; -(**2**), AgmF reaction without substrate **2** added. **e** Proposed enzymatic mechanism for the AgmF-catalyzed reaction.

**AgmE, C, and B harbor individual activities of APRTase, ribose 5-P pyrophosphokinase, and AMP phosphatase.** An initial in silico analysis showed that AgmE contains a conserved phosphoribosyltransferase domain. To evaluate if the protein carries out the expected adenine phosphoribosyltransferase (APRTase) function, it was overexpressed and purified from *E. coli* BL21(DE3),

and assayed in vitro with the substrates adenine and phosphoribosyl pyrophosphate (PRPP). HPLC analysis of the reaction products showed that the AgmE-catalyzed reaction is capable of generating a characteristic AMP peak, whose identity was further verified by LC-MS, and indicating that AgmE plays a similar role as APRTase during **1** biosynthesis (Supplementary Fig. 17).

As for the catalytic function of AgmC, in silico analysis indicated that it harbors a conserved ribose-phosphate pyrophosphokinase domain. To determine if this enzyme executes a related activity, we performed an in vitro enzyme assay using ribose 5-phosphate (ribose 5-P) as substrate. To facilitate the detection of the enzymatic product (PRPP), we conducted an assay to indirectly monitor the conversion of ATP to AMP, using a coupled AgmC-AgmE reaction to trap PRPP to form the product AMP. As anticipated, the coupled reaction synthesized an enhanced amount of AMP in comparison with that of the negative control (without adenine added), showing that AgmC could utilize ATP to give AMP (Supplementary Fig. 18d). The AgmC-AgmE-coupled reactions lacking ribose 5-P (or lacking both ribose 5-P and ATP) also abolished the production of the target product AMP. These data demonstrate that AgmC performs a catalytic pyrophosphokinase-like function in the process of 1 biosynthesis (Supplementary Fig. 18).

Similar bioinformatics analysis for the function of AgmB revealed a conserved HAD signature motif with 51% identity to C5D20_10175 (HAD-like hydrolase) of *Rathayibacter toxicus*. This family of enzymes unusually catalyzes nucleophilic substitution reactions at phosphorus or carbon centers, using a conserved Asp carboxylate anion in the active site[40]. All members of this family possess a conserved alpha/beta core domain. To assess the function of the purified AgmB enzyme, we tested its activity in vitro and, as expected, found that the AgmB enzyme is indeed able to convert AMP to adenosine, the identity of which was verified by LC-MS (Supplementary Fig. 19). Collectively, this provides biochemical evidence that AgmE, AgmB, and AgmC have the activities of APRTase, AMP phosphatase, and ribose 5-P pyrophosphokinase, and they are most likely to play related functional roles during 1 biosynthesis.

## Reconstitution of the complete six-enzyme pathway in vitro to 1 and 2.

The functional role of AgmD in the 1 biosynthetic pathway also remained undetermined, and by in silico analysis showed 31% identity to AlsE of *E. coli*. The AlsE enzyme was previously characterized as a D-allulose 6-phosphate 3-epimerase that utilizes D-fructose 6-phosphate as substrate. In addition, earlier metabolic labeling experiments indicated that 1 arises directly from either D-glucose or D-fructose. This suggested that AgmD should play a 3-epimerase role similar to AlsE and provides the basis for further understanding of 1 biosynthesis (Supplementary Fig. 20a–c).

To further decipher the biosynthetic logic underlying 1 biosynthesis, we reconstituted the complete biosynthetic pathway to 1 in vitro. The six enzymes covering AgmA-F were individually overexpressed and purified from *E. coli* BL21(DE3), allowing for the construction of the biosynthetic pathway to 1 de novo (Fig. 5a, b). HPLC analysis of the products showed that the complete AgmA-F reactions are capable of producing the distinctive 2, adenine, and 1 peaks, whose corresponding $[M + H]^+$ ions $m/z$ 298.1143, 136.0617, and 280.1039 (and related fragment ions) are fully consistent to those of authentic standards. It was also found that the enzymatic reaction including AlsE, AgmA-C, and AgmE-F is also capable of producing the peaks of 2, adenine, and 1 peaks, but with a higher overall efficiency, showing that AlsE and AgmD share the same functional role. These data demonstrated that the six enzymes (AgmD, C, A, E, B, and F) catalyze the relay reactions with fructose 6-phosphate as the starter substrate in the biosynthesis of angustmycins (Fig. 5c and Supplementary Table 5).

Based on these results, we then attempted to dissect the individual stepwise reactions occurring during 1 biosynthesis. HPLC analysis of the products indicated that the reaction (AlsE + AgmC) was able to generate a characteristic AMP peak, which was absent from the AlsE reaction. LC-MS analysis of this peak gave a $[M + H]^+$ molecular ion at $m/z$ 348.0703 and major fragment ions at 136.0617 and 250.0937, fully matched to those of the AMP authentic standard. These data established that AgmC is a pyrophosphokinase that uses allulose 6-phosphate as the acceptor substrate. The coupled reactions containing AlsE, AgmC, and AgmA also gave the adenine peak, but without the coproduction of AMP. Hence, the substrate of AgmA is probably also the product of AgmC. This intermediate was further consumed by the addition of the AgmE enzyme in the related coupled reaction to form 5. The coupled LC-MS analysis of 5 gave a $[M + H]^+$ ion at $m/z$ 378.0806, and main fragment ions of $m/z$ 360.0691, 262.0933, and 136.0619, demonstrating that AgmE is an adenine phosphoallulosyltransferase. 5 was revealed to be specifically dephosphorylated to render 2, whose identity was confirmed by LC-MS (Fig. 5c, Supplementary Fig. 21, and Supplementary Table 5). To analyze their relative catalytic activities, the generation of 2 by the five-enzyme coupled reaction was measured with either AlsE or AgmD. The relative activity of AgmD is 22% when compared with that of AlsE, consistent with the result of the one-pot in vitro reaction (Supplementary Fig. 20d). The stepwise reactions were also individually carried out to illustrate that AgmD has a function identical to AlsE (Supplementary Fig. 22 and Supplementary Table 5), and the accumulation of adenine and consumption of 5 would affect the one-pot reaction efficiency (Supplementary Figs. 23 and 24).

In the present paper, we have demonstrated that the complete 1 biosynthetic pathway consisting of six enzymes is sequentially arranged to synthesize 1. The biosynthetic pathway starts with a glycosyl epimerization step from D-fructose 6-phosphate to D-allulose 6-phosphate as catalyzed by AgmD, and then successively undergoes pyrophosphorylation, adenine phosphoallulosyltransfer, and dephosphorylation to produce 2. After the final dehydration step, 2 is converted to 1. Notably, the cofactors involving $NAD^+$ and $NADP^+$ are recycled during 1 biosynthesis, and AMP plays dual roles as both the substrate and the product. These latter results suggest that 1 biosynthesis exploits a self-sufficient and efficient strategy for cofactor recycling and substrate utilization, which is unusual in natural product biosynthesis (Fig. 5d).

## Utilization of *E. coli* as a cell factory for engineered production of 1 and 2.

Finally, we have also explored the potential of utilizing *E. coli* as a cell factory for the production of 1. The six biosynthetic genes (*agmDCAEBF*, listed according to the reaction sequence) were introduced into *E. coli* GYJ23, a strain engineered for nucleoside production, via a three-plasmid system (Fig. 6a). After confirmation, the transformant GYJ23/*agmDCAEBF* was shown by HPLC to be capable of generating the target peaks of 1 and 2, whose identities were further confirmed by LC-MS analysis (Fig. 6b and Supplementary Table 5). The yields of 2 and 1 could independently reach 110 and 370 μg/mL after 96 h fermentation. Interestingly, we found the strain (GYJ23/*agmDCAEB*, lacking *agmF*) is capable of targeted accumulating 2 with a yield of 420 μg/mL after 96 h fermentation (Fig. 6c, d).

We also evaluated the production of 1 and 2 conferred by the set of six genes with *agmD* replaced by *alsE*. HPLC analysis showed that the strain GYJ23/(*alsE* + *agmCAEBF*) also produces 1 and 2, but with higher titers (Fig. 6b and Supplementary Table 5). Hence, the 2 titer of the strain GYJ23/(*alsE* + *agmCAEB*) was 780 μg/mL, which is considerably higher than that of the GYJ23/*agmDCAEB* strain. All of these data demonstrate that suitably engineered *E. coli* strains could be used as a robust cell factory for the efficient production of the nucleoside antibiotic 1

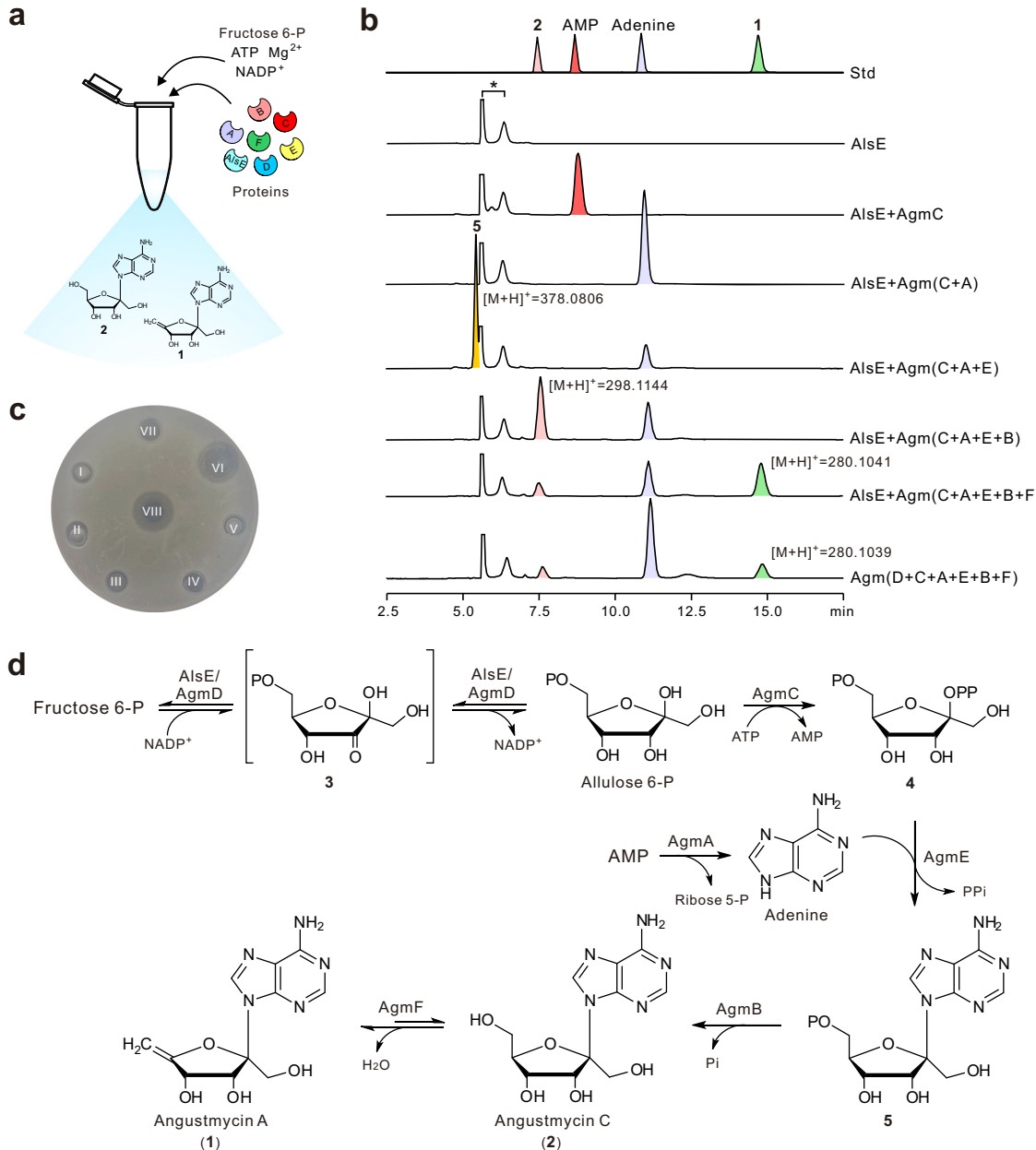

**Fig. 5 Reconstitution of the whole 1 biosynthetic pathway in vitro. a** Diagrammatic sketch of one-pot reaction in vitro for **2** and **1** biosynthesis. **b** HPLC analysis ($\lambda = 254$ nm) of the one-pot reaction mixtures of proteins responsible for angustmycin biosynthesis. Std, the authentic standards of **2**, AMP, adenine, and **1**; AlsE, reaction with AlsE added, accompanying with fructose 6-P, metal ion, and cofactors; Agm(D + C + A + E + B + F), reaction with complete six proteins (AgmD, AgmC, AgmA, AgmE, AgmB, and AgmF) added, accompanying with related substrates, metal ion, and cofactors. Other samples are correspondingly assigned. '*' represents NADP+-related compounds. **c** Bioassays of the **1** authentic standard and the products generated by related reaction mixtures of different protein compositions. The indicator strain is *Mycobacterium smegmatis* mc²155. The Roman numerals I, II, III, IV, V, VI, VII, and VIII represent AlsE, AlsE + AgmC, AlsE + Agm(C + A), AlsE + Agm(C + A + E), AlsE + Agm(C + A + E + B), AlsE + Agm(C + A + E + B + F), Agm(D + C + A + E + B + F), and **1** standard, respectively, and the denotations of them correspond to those shown in (**b**). **d** Confirmed biosynthetic pathway to **2** and **1** on the basis of one-pot reaction. Fructose 6-P, Fructose 6-phosphate.

(Fig. 6c, d). The successful production of **1** and **2** in a heterologous *E. coli* cell factory raised the question of why the strain GYJ23/(*alsE* + *agmCAE*) is not capable of accumulating **5** and adenine. This may in part be because phosphate-containing compounds, such as nucleotides, are less easily transported out of cells. Concerning adenine, as a primary metabolite, it is also perhaps prone to being recycled during the cell cycle of *E. coli*.

In summary, we report the discovery and characterization of the gene cluster responsible for angustmycin A (**1**) biosynthesis

and we provide evidence that AgmA (phosphoribohydrolase), AgmB (phosphatase), AgmC (pyrophosphokinase), AgmD (epimerase), and AgmE (phosphoallulosyltransferase) collaborate together to biosynthesize the angustmycin C (**2**) molecule. We have also unraveled that AgmF functions as an unusual dehydratase for the final tailoring step to **1** via a self-sufficient strategy for NAD+ cofactor recycling. Moreover, we have reconstituted the complete biosynthetic pathway to **1** in vitro and successfully engineered the production of **1** in a robust cell

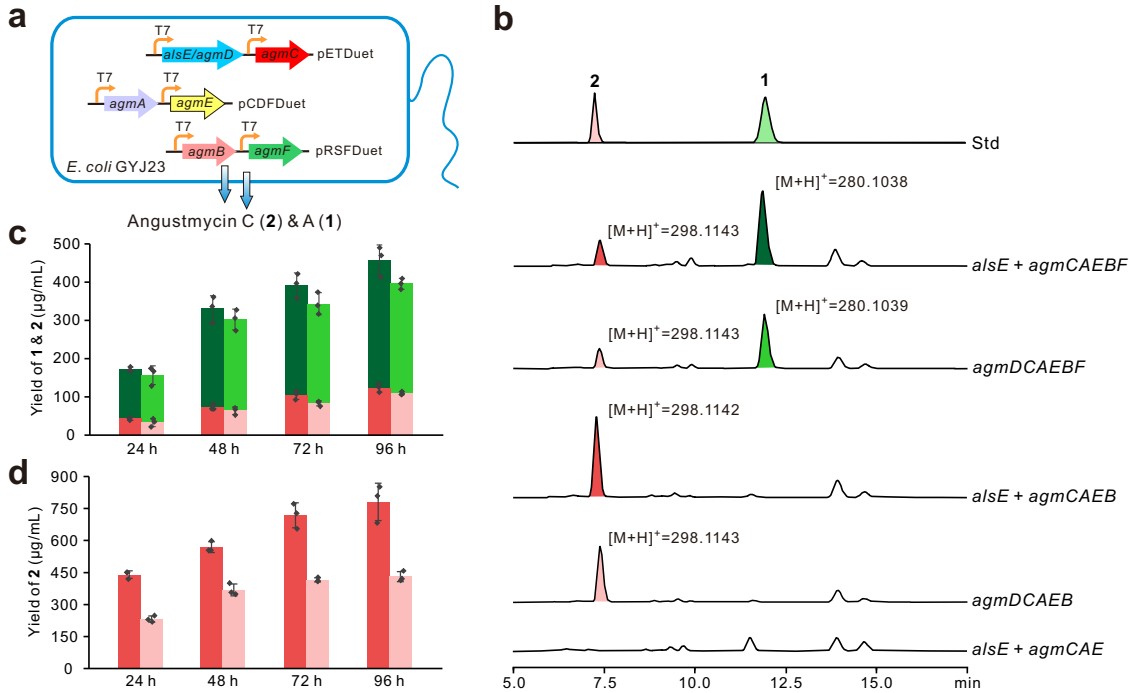

**Fig. 6 Engineered production of 2 and 1 in *Escherichia coli*. a** Diagrammatic sketch of the production of **2** and **1** in *E. coli*, the biosynthesis-related genes are cloned into pETDuet, pCDFDuet, and pRSFDuet and coexpressed in *E. coli* GYJ23. **b** HPLC analysis ($\lambda = 254$ nm) of metabolites producedby related recombinants of *E. coli* GYJ23. Std, the authentic standards of **2** and **1**; *alsE* + agm*CAEBF*, the metabolites of the recombinant *E. coli* GYJ23 containing pETDuet/*alsE* + *agmC*, pCDFDuet/*agmA* + *agmE*, and pRSFDuet/*agmB* + *agmF*. *agmD* + agm*CAEBF*, the metabolites of the recombinant *E. coli* GYJ23 containing pETDuet/*agmD* + *agmC*, pCDFDuet/*agmA* + *agmE*, and pRSFDuet/*agmB* + *agmF*. Other samples are correspondingly assigned. **c** The production of **2** and **1** by the strains GYJ23/(*alsE* + *agmCAEBF*) and GYJ23/*agmDCAEBF*. The red bars refer to **2** production, while the green ones refer to **1** production, and the colors marked correspond to those in HPLC analysis of (*alsE* + *agmCAEBF*) and *agmDCAEBF* in (**b**). **d** The production of **2** by fermenting strain GYJ23/(*alsE* + *agmCAEB*) and GYJ23/*agmDCAEB*. The 24, 48, 72, and 96 h represent the corresponding time for fermentation sampling, and the values are the means ± s.d. measured from three biological replicates, and the colors marked correspond to those in HPLC analysis of (*alsE* + *agmCAEB*) and *agmDCAEB* in (**b**). Source data underlying (**c**) and (**d**) are provided as a Source Data file.

factory of *E. coli*. We anticipate that deciphering the angustmycin pathway will expand the biochemical repertoire for the intriguing enzymatic reactions leading to nucleoside natural product biosynthesis (while this paper was under review, a part result of the angustmycin biosynthesis was reported by Shiraishi et al.[41]; they characterized the gene cluster and identified part of the AgmF function (dehydratase)), and open the way to the rapid and rational discovery of more purine nucleoside antibiotics related to **1** (Supplementary Fig. 25).

## Methods

**General methods**. Strains and plasmids used in this study are described in Supplementary Table 6 and primers are listed in Supplementary Table 7. General methods employed in this work follow the standard protocols[42,43].

**Enzymes, chemicals, and reagents**. All of the enzymes used in this study were purchased from New England Biolabs. The chemicals and reagents were the products of Sigma-Aldrich, Thermo Scientific, OMEGA, or J&K Scientific.

**Sequencing analysis of the genome of actinobacteria strain**. Sequencing of the genome of the actinobacteria strain was performed on Nanopore and Illumina Hiseq2500 machine. The sequencing reads for *S. angustmyceticus* JCM 4053 and *S. decoyicus* NRRL 2666 are 7.41 and 7.97 million, respectively. The raw data were assembled by Flye (Version: 2.8.3) and Unicycler (Version: 0.4.9) to obtain the scaffold. Hmmer (Version: 3.3.1) software was used for the annotation of the genomes. Bioinformatic analysis of the target DNA region was performed on the basis of the online programs FramePlot 4.0beta (http://nocardia.nih.go.jp/fp4/) and 2ndFind (http://biosyn.nih.go.jp/2ndfind/).

**Direct cloning of the whole *agm* gene cluster and its variants**. The whole *agm* gene cluster was directly cloned by a two-step PCR strategy and the detailed procedure is described as follows: the two pieces (left, 4603 bp; right, 4636 bp) housing the whole

*agm* gene cluster were individually amplified by PrimeSTAR Max DNA Polymerase (Takara) with primers (piece1-F/R and piece2-F/R) (Supplementary Table 7) and template (genomic DNA of *S. angustmyceticus* JCM 4053) (Supplementary Fig. 2a). The two pieces were cloned into the site between *Eco*RI and *Xba*I of the vector pSET152 by Gibson Assembly method to obtain the resulting plasmid pCHW501 (Supplementary Fig. 2a). The *agmF*-deleted gene cluster cloning was similar to that described above. The two pieces (left, 4603 bp; right, 3148 bp) were individually amplified by PrimeSTAR Max DNA Polymerase (Takara) with primers (piece1-F/R and piece3-F/R) (Supplementary Table 7) and template (genomic DNA of *S. angustmyceticus* JCM 4053) (Supplementary Fig. 2b). The two pieces were cloned into the sites between *Eco*RI and *Xba*I of vector pSET152 by Gibson Assembly method to obtain the plasmid pCHW501Δ*agmF* (Supplementary Fig. 2b). The two recombinant plasmids were confirmed by PCR with primers agm id-F/R (Supplementary Table 7). The *agmR*-deleted gene cluster cloning was based on pCHW501. Two pieces (left, 3042 bp; right, 417 bp) were individually amplified by PrimeSTAR Max DNA Polymerase (Takara) with primers (DagmR-First-F/R and DagmR-Second-F/R) (Supplementary Table 7) and template (genomic DNA of *S. angustmyceticus* JCM 4053). The two pieces were cloned into the site between *Eco*RI and *Bgl*II of vector pCHW501 by Gibson Assembly method to obtain the *agmR* in-frame deletion plasmid pCHW501Δ*agmR*, which was confirmed by PCR with primers agmR id-F/R (Supplementary Fig. 2c and Supplementary Table 7).

**Fermentation and detection of angustmycins**. For fermentation, *S. angustmyceticus* JCM 4053 and *S. decoyicus* NRRL 2666 were independently cultivated on YS agar (including 2 g yeast extract, 10 g soluble starch, and 15 g agar per liter, pH 7.3) and MS plate[43], and spores of related strains were inoculated in ISP2 medium[44] and cultivated for 2 days; after that, the cultures (2%, V/V) were transferred to fermentation medium[45] and fermented (180 r/min, 30 °C) for 5 days. For HPLC and LC-high resolution mass spectrometry (HRMS) analysis, the fermentation broth was processed by adding oxalic acid till pH 5.0. HPLC analysis was performed using Shimadzu LC-20AT equipped with C18 column (Diamonsil, 5 μm, 4.6 × 250 mm) under 0.15% aqueous formic acid (95%): methanol (5%) at a flow rate of 0.5 mL/min over 30 min and the detection wavelength was 254 nm. LC-HRMS analysis was carried out on a Thermo Fisher Scientific ESI-LTQ Orbitrap (Scientific Inc.) instrument controlled by Xcalibur in the positive mode. The

parameters for HRMS analysis are as follows: a capillary temperature of 275 °C and a capillary voltage of 35 V, and the error in this work is 10 p.p.m. The bioassay method was described in Supplementary Method 1, and the extraction, purification, and chemical structure identification of angustmycins were described in Supplementary Methods 2 and 3.

**Model building and analysis of AgmF structure**. The homology structural model of AgmF was constructed according to the X-ray structure of NP_217765 from *Mycobacterium tuberculosis* H37Rv (PDB 3CE6, https://www.rcsb.org/structure/3CE6) and further refined using Discovery Studio 3.0 (Accelrys).

**In vitro assay of AgmA**. Reaction mixtures (50 µL each) consisted of a solution containing 50 mM Tris-Cl buffer (pH 7.5), 1 mM AMP (dAMP), and 10 µg AgmA at 30 °C. The reaction was terminated after 2 h by the addition of an equivalent volume (50 µL) of methanol. Following centrifugation to remove the protein, the reaction was analyzed by HPLC and LC-HRMS equiped with a reverse-phase C18 column with 0.15% aqueous formic acid (95%): methanol (5%) at a flow rate of 0.5 mL/min over 30 min, and the detection wavelength was 254 nm.

**In vitro assay of AgmF**. Reaction mixtures (50 µL each) consisted of a solution containing 50 mM PBS buffer (pH 7.5), 1 mM NAD$^+$, 1 mM **2**, and 10 µg AgmF at 30 °C. The reaction was terminated after 2 h by the addition of an equivalent volume (50 µL) of methanol. Following centrifugation to remove the protein, the reaction was analyzed by HPLC and LC-HRMS equiped with a reverse-phase C18 column with 0.15% aqueous formic acid (95%): methanol (5%) at a flow rate of 0.5 mL/min over 30 min and the detection wavelength was 254 nm.

**In vitro assay of AgmB, AgmC, and AgmE**. AgmB reaction mixtures (50 µL each) consisted of a solution containing 50 mM Tris-Cl buffer (pH 7.5), 1 mM AMP, 1 mM Mg$^{2+}$, and 10 µg AgmB at 30 °C for 2 h; AgmE reaction mixtures (50 µL each) were composed of a solution containing 50 mM PBS buffer (pH 7.5), 1 mM adenosine, 1 mM PRPP, 1 mM Mg$^{2+}$, and 10 µg AgmE at 30 °C for 2 h; AgmC and AgmE combined reaction mixtures (50 µL each) consisted of a solution containing 50 mM Tris-Cl buffer (pH 7.5), 1 mM ribose 5-P, 1 mM ATP, 1 mM Mg$^{2+}$, 10 µg AgmC, and 10 µg AgmE at 30 °C for 2 h. The post-processing and analysis methods of these reactions were identical to that of the AgmF reaction. The protocol of overexpression and purification of the target proteins was described in Supplementary Method 4.

**One-pot reaction for 1 and 2 biosynthesis**. Six-protein reaction mixtures (50 µL each) consisted of a solution containing 50 mM Tris-Cl buffer (pH 8.0), 10 mM fructose 6-phosphate, 1 mM ATP, 0.5 mM NADP$^+$, 4 mM Mg$^{2+}$, and 10 µg of individual proteins (AgmD/AlsE, AgmC, AgmA, AgmE, AgmB, and AgmF) at 30 °C for 2 h. The reaction was terminated after 2 h by the addition of an equivalent volume (50 µL) of methanol. Following centrifugation to remove the protein, the reaction was analyzed by HPLC and LC-HRMS equiped with a reverse-phase C18 column with 0.15% aqueous trifluoroacetic acid (95%): methanol (5%) at a flow rate of 0.5 mL/min over 30 min. The process of heterologous expression in *E. coli* was as described in Supplementary Method 5.

**Reporting summary**. Further information on research design is available in the Nature Research Reporting Summary linked to this article.

## Data availability

Genomes of *S. angustmyceticus* JCM 4053 and *S. decoyicus* NRRL 2666 have been deposited in GenBank under accession nos. CP082945 and CP082301, respectively. The two gene cluster sequences from *S. angustmyceticus* JCM 4053 and *S. decoyicus* NRRL 2666 have been deposited in GenBank under accession nos. MZ151497 and MZ151498, respectively. All other data generated and analyzed in this study are available within the article and the Supplementary information. Source data are provided with this paper.

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

## Acknowledgements

This work was supported by grants National Key R & D Program of China (2021YFC2100600 to W.C.), the National Natural Science Foundation of China (31770041, 31970052, and 32170026 to W.C.), the Hubei Province's Outstanding Medical Academic Leader program to W.C., and the Open Funding Project of the State Key Laboratory of Microbial Metabolism (MMLKF19- 03 to W.C.). Mention of any trade names or commercial products in this publication is solely for the purpose of providing specific information and does not imply recommendation or endorsement by the U.S. Department of Agriculture, who is an equal opportunity provider and employer.

## Author contributions

L.Y., W.Z., and Y.S. conducted the genetic and biochemical experiments. Y.-S.C. analyzed the NMR data. H.M. helped with the biochemical experiments. M.J. assisted with the proposing of the related enzymatic mechanism. W.C. and N.P.J.P conceived and directed the research. W.C. and N.P.J.P. wrote the manuscript. Z.D. revised the manuscript.

## Competing interests

The authors declare no competing interests.
