## [Peer Review File · Nature Communications]

Efficient Biosynthesis of Nucleoside Cytokinin Angustmycin A Containing an Unusual Sugar SystemReviewers' Comments:

Reviewer #1:

Remarks to the Author:

This is a very interesting study from the Price and Chen groups that detail the biosynthesis of a new nucleoside analog that is identified in actinobacteria that contains an unusual sugar system. The group used a genome-mining approach to analyze the whole genomes sequences of two producers of angustmycin A (AGM-A) that contains an unusual C5'-C6' ene containing sugar linked to adenine. The group demonstrated the viability of the purported biosynthetic gene cluster using deletion knockouts and identify a gene that catalyzes a dehydration step to afford the dehydro species. Overall the work is solid and could be suitable for publication in Nature Comm. but there are still significant issues that need to be addressed.

Major item of concern: the authors identify AgmF, an analog of SAH hydrolase (SAHH) as the enzyme responsible for catalyzing the dehydration of AGM-C to produce AGM-A. However, there are concerns about the mechanism shown in Figure 4e and the authors analysis of the function of this protein. The mechanism differs from the canonical SAHH mechanism in that a tautomerization step is proposed to occur after the first oxidation. In canonical SAHs, there is a proton abstraction at C4' that is catalyzed by Glu130 (rat liver enzyme numbering). However, the equivalent Glu129 is present in the sequence of AgmF and the authors show that Ala substitution at this residue abolishes activity (Figure 4D, 4th trace). Hence, the mechanism as drawn can not be correct. The authors should provide evidence of the divergence from canonical SAHH mechanistic, or change this mechanism

Secondy, the authors propose the reaction from 1b to 1c involves elimination of hydroxide. In canonical SAHH, the leaving group (homocysteine) is an activated species (protonated by a His56 equivalent). Please redraw this step so that it makes chemical sense.

Minor points:

1. I don't understand the point of Figure 3b. This should really be shown as a tree rather than an SSN if the point is to show the phylogeny of these genes.
2. The authors should include an SSN of AgmF homologs, and see how the biosynthetic genes might differ from the metabolic genes.
3. The authors state that adenosine is an inhibitor of AgmF catalyzed formation of AGM-A. Is adenosine also a substrate?
4. Please comment on whether the k_{cat}/K_m values agree with those reported for the biosynthetic SAHH.
5. The story would really benefit from a crystal structure but in the absence of this, can the authors rationalize why SAH is not a substrate? There must be mutations that block the active site.
6. From the reconstitution experiments shown in Figure 5c, it looks like adenine is a competitive inhibitor of at least one enzyme past AgmA. Can the authors comment?

Reviewer #2:

Remarks to the Author:

Yu et al. report the full biosynthetic pathway of Angustmycin A (AGM-A), an adenosine analog containing a rare β -2-N-ketoglycosidic bond. AGM antibiotics have been studied for over fifty years with documented work on inhibition of GMP synthesis in Gram positive bacteria, anti-mycobacterial,

and cytokinin activities. Despite extensive studies on the biochemical activities of AGM antibiotics, their biosynthesis has remained elusive. In the current comprehensive work, Yu et al. identified a putative biosynthetic gene cluster through genome mining, confirmed its relevance through heterologous expression, and further reconstituted the six-enzyme pathway both in vitro and in *E. coli* to generate AGM-A. Although all identified enzymes' activity can be readily predicted based on bioinformatic analysis, the overall pathway seems interesting, exploiting a quite self-sufficient and efficient strategy for cofactor recycle and substrate utilization. In addition, the relatively high-titer of AGM-A and AGM-C production in *E. coli* (close to g/L) suggests potential future application for these nucleoside antibiotic production and engineering. The overall conclusions are mostly supported by data, and the results will benefit the natural product biosynthesis field, in particular nucleoside antibiotic biosynthesis. However, the manuscript needs to significantly improve before publication. Here are some specific comments:

Major:

1. On AgmF, figure S10 makes the cofactor requirement of NAD very confusing. Is there any evidence showing reduction of NAD+? Line 251, what is the meaning of maintaining activity without addition of NAD? Maybe some time-course experiments with and without NAD or NADH can be performed to further scrutinize reaction equilibrium. Figure S13a is quite confusing. The legend says substrate concentration ranging 0.9-10 mM which is inconsistent with data in the figure. In addition, the inhibition from adenosine needs to be better and quantitatively characterized to claim competitive inhibition.
2. Please also show the biochemical activity of AgmC in converting ATP to AMP if PRPP cannot be readily detected. The current coupled assay results with AgmE are somewhat weak.
3. The kinetics/relative activities of AlsE and AgmD should be reported.

Minor:

1. In Introduction, AGM-A and C "interconvertible" or "interconverted" appeared a couple of times, but the evidence for this hypothesis is lacking.
2. It is suggested to consolidate and briefly discuss the most important bioactivity of AGM antibiotics in one paragraph. This is not the emphasis of the current manuscript and the current discussion is a bit too long.
3. Line 176, it is premature to state that agmA-F are essential for AGM biosynthesis here. What is the genetic evidence?
4. Line 191, this implication is too specific on regulating transcriptional level of agmF. Consider revision.
5. Line 242-244, what is the logic here? The previous heterologous expression has suggested AgmF converts AGM-C to A.
6. Many initial biochemical analysis of AgmE, C and B used alternative substrates, rather than proposed ones in AGM biosynthesis. These data can suggest possible roles of these enzymes in AGM biosynthesis but not definitive. Please tone down relevant discussions.
7. All figures with LC traces need to mention the detection method in figures or legends. What are the signals shown in all figures?
8. PPM errors for all HRMS data need to be reported.
9. The scholarly presentation of manuscript needs to improve. There are numerous non-standard English and grammar and other mistakes. For example, line 253, Figure 2a. Line 260, Figure 2b.
10. The TOC to start with AMP seems not appropriate. ATP should be used.

Reviewer #3:

Remarks to the Author:

The manuscript by Yu et al describes the discovery of the Angustmycin biosynthetic pathway from *Streptomyces angustmyceticus*. As a nucleoside antibiotic produced by *Streptomyces*, Angusmycin A (AGM-A) has interesting cytokinin activities and is structurally distinctive. The glycosyl unit of AGM-A is a unique six-carbon ketose sugar (β -D-psicofuranosyl) with C5'-C6' dehydration. This sugar is linked to adenine via an unusual β -2-N-ketoglycosidic bond. Among the numerous nucleoside pathways,

there are no examples that can provide direct clues to the biosynthesis of AGMs. In this work, the authors proposed the target biosynthetic gene cluster of AGM (*agm*) through comparative genomics strategy, and confirmed it through heterologous expression and gene deletion experiments. The detailed in vitro experimental data including one-pot biosynthesis provided in this manuscript convinced me that AgmA-E catalyzes the biosynthesis of AGM-C, and AgmF catalyzes the dehydration in the last step to produce AGM-A. In addition, they reconstructed the complete six-enzyme pathway in *E. coli* and achieved high-yield production of AGM-A, although this is not the core goal of this study. In general, I think this work has solved a new type of nucleoside antibiotic biosynthesis pathway, in which the interesting steps include the formation of the unique β -2-N-keto glycosidic bond catalyzed by AgmE and the atypical dehydration process catalyzed by AgmF.

This finding expands the knowledge of nucleoside skeleton formation and dehydration mechanisms in the field of biosynthesis and the work will be of great interest to the readers of Nature Communications. While, the manuscript still needs to be improved:

-- The AGM pathway proposed by the author is based on the results of one-pot enzymatic reactions in vitro. However, in a step-by-step in vitro experiment on AgmC-E-B, the starting substrate was Ribose-5-P, not Allulose-6-P in the AGM pathway. In addition, it is speculated that AgmD catalyzes Fructose-6-P to Allulose-6-P in the AGM pathway, which lacks direct evidence support (the substitution of AgmD with a homologous AlsE in the 6-enzyme one-pot reaction is not rigorous enough). Therefore, in order to solve the above problems, I think at least the following experiments need to be done:

- 1) One-pot reaction of AgmD and AgmC with Fructose-6-P as a starting compound, by detecting AMP production to determine whether Fructose-6-P can be used as a substrate of AgmD.
- 2) One-pot reaction of AgmD, AgmC, AgmE and AgmA with Fructose-6-P as a starting compound, by detecting compound 4 production to determine whether AgmE can catalyze the formation of the key glycosidic bond in AGMs.

Minor comments:

-Picture S10 is wrong. This picture is the same as Picture S9.

-In the manuscript or SI, there is no information about the detection wavelength for HPLC analysis.

-Line 138, Please specify which gene cluster was cloned.

-Line 143, M1154::pCHW501 repeated.

-Line 178-180, The information of these two less critical proteins has been included in Table S1 and does not need to be described in the text.

-Line 211, dAMP \diamond dADP

-Line 253, Figure 2a \diamond Figure 4a (?)

-Line 259, Figure 2b \diamond Figure 4b (?)

-Line 268, "Mycobacterium tuberculosis H37Rv (PDB: 3CE6)", is this SAH hydrolase a NAD⁺ self-sufficient enzyme like AgmF? The authors can discuss it from its catalytic mechanism and structural similarity.

-Line 305 and Figure S14, the phylogenetic analysis of AgmF and homologues does not include the structural model protein (Rv3248c from Mycobacterium tuberculosis H37Rv, PDB: 3CE6)? Which branch of the phylogenetic tree is Rv3248c located?

-Line 326-330 and Figure S16d, the negative controls should include reactions without AgmC or

AgmE.

-Figure S19, NADH+ \diamond NADP+

Response to Reviewer 1

Q1: The authors identify AgmF, an analog of SAH hydrolase (SAHH) as the enzyme responsible for catalyzing the dehydration of AGM-C to produce AGM-A. However, there are concerns about the mechanism shown in Figure 4e and the authors analysis of the function of this protein. The mechanism differs from the canonical SAHH mechanism in that a tautomerization step is proposed to occur after the first oxidation. In canonical SAHs, there is a proton abstraction at C4' that is catalyzed by Glu130 (rat liver enzyme numbering). However, the equivalent Glu129 is present in the sequence of AgmF and the authors show that Ala substitution at this residue abolishes activity (Figure 4D, 4th trace). Hence, the mechanism as drawn can not be correct. The authors should provide evidence of the divergence from canonical SAHH mechanistic, or change this mechanism.

A1: Many thanks for your constructive suggestions, and we have made related modifications for the enzymatic mechanism of AgmF (which has been updated in Figure 4e) in the revised manuscript.

Q2: The authors propose the reaction from 1b to 1c involves elimination of hydroxide. In canonical SAHH, the leaving group (homocysteine) is an activated species (protonated by a His56 equivalent). Please redraw this step so that it makes chemical sense.

A2: Many thanks for your kind reminder, and we have already redrawn the enzymatic step in the revised manuscript.

Q3: I don't understand the point of Figure 3b. This should really be shown as a tree rather than an SSN if the point is to show the phylogeny of these genes.

A3: Thanks for your kind comments, actually, the SSN analysis of AgmA was used to show the information that the cytokinin-activating family proteins are widespread in bacteria genomes, as a result, it would be better if we could keep it in the present manuscript.

Q4: The authors should include an SSN of AgmF homologs, and see how the biosynthetic genes might differ from the metabolic genes.

A4: Thanks for your constructive suggestions, and we have added the SSN analysis of AgmF homologs in the supplemental information of the revised manuscript (Figure S16).

Q5: The authors state that adenosine is an inhibitor of AgmF catalyzed formation of AGM-A. Is adenosine also a substrate?

A5: Thanks for your kind reminder, and actually adenosine is not a substrate of AgmF (related data was indicated in Figure S12e).

Q6: Please comment on whether the k_{cat}/K_m values agree with those reported for the biosynthetic SAHH.

A6: Commented as suggested.

Q7: The story would really benefit from a crystal structure but in the absence of this, can the authors rationalize why SAH is not a substrate? There must be mutations that block the active site.

A7: Many thanks for your kind suggestion and the work on the crystal structure of AgmF are underway in our laboratory, and we do expect to solve it in the future.

Q8: From the reconstitution experiments shown in Figure 5c, it looks like adenine is a competitive inhibitor of at least one enzyme past AgmA. Can the authors comment?

A8: Many thanks for your constructive comments, and the accumulated adenine in the one pot reaction is mainly due to the following reasons:

(i) AgmA is also capable of recognizing compound **4** as substrate to generate adenine and allulose 6-P (Figure S24), and the latter is further catalyzed by AgmC to form compound **3** with consuming of ATP.

(ii) The byproduct ribose 5-P can also be activated to PRPP by AgmC with consuming

of ATP (Figure S18).

(iii) The accumulation of adenine is capable of competitively inhibiting the AgmF activity (Figure S23).

For these reasons, the molecular ratio of adenine and compound **3** in the reactions is far more than 1.0, which leads to the accumulation of adenine in the one pot reaction.

Response to Reviewer 2

Q1: On AgmF, figure S10 makes the cofactor requirement of NAD very confusing. Is there any evidence showing reduction of NAD+?

A1: Sorry for this and we found that Figure S10 was previously missed by accident, and the correct one has already been added in the revised manuscript.

Q2: Line 251, what is the meaning of maintaining activity without addition of NAD? Maybe some time-course experiments with and without NAD or NADH can be performed to further scrutinize reaction equilibrium.

A2: Many thanks for your suggestion, and we have conducted the time-course experiment, and related data has been added in the revised version of the manuscript (Figure S10).

Q3: Figure S13a is quite confusing. The legend says substrate concentration ranging 0.9-10 mM which is inconsistent with data in the figure.

A3: Sorry for this, and we have corrected the legend of Figure S13a to make it more understandable in the revised manuscript.

Q4: The inhibition from adenosine needs to be better and quantitatively characterized to claim competitive inhibition.

A4: Many thanks for your kind suggestion, and we have supplemented the

experimental data in Figure S14.

Q5: Please also show the biochemical activity of AgmC in converting ATP to AMP if PRPP cannot be readily detected. The current coupled assay results with AgmE are somewhat weak.

A6: Thanks for your kind suggestion, and the biochemical activity of AgmC has been added in the supplemental Figure S18.

Q7: The kinetics/relative activities of AlsE and AgmD should be reported.

A7: Thanks for your valuable suggestions, and we have conducted additional experiment to compare the relative activity of AlsE and AgmD (Figure S20).

Q8: In Introduction, AGM-A and C “interconvertible” or “interconverted” appeared a couple of times, but the evidence for this hypothesis is lacking.

A8: Many thanks for your kind suggestion, and we have removed related sentences containing the words “interconvertible” and “interconverted” to avoid potential misunderstandings.

Q9: It is suggested to consolidate and briefly discuss the most important bioactivity of AGM antibiotics in one paragraph. This is not the emphasis of the current manuscript and the current discussion is a bit too long.

A9: Thank you for your constructive suggestion, and we have made related modifications in the revised manuscript.

Q10: Line 176, it is premature to state that agmA-F are essential for AGM biosynthesis here. What is the genetic evidence?

A10: Sorry for this and we have modified the description to make it clearer to readers.

Q11: Line 191, this implication is too specific on regulating transcriptional level of agmF. Consider revision.

A11: Revised as suggested.

Q12: Line 242-244, what is the logic here? The previous heterologous expression has suggested AgmF converts AGM-C to A.

A12: Sorry for this and we have made related modifications in the revised manuscript.

Q13: Many initial biochemical analysis of AgmE, C and B used alternative substrates, rather than proposed ones in AGM biosynthesis. These data can suggest possible roles of these enzymes in AGM biosynthesis but not definitive. Please tone down relevant discussions.

A13: Thanks for your valuable comments, and we have modified related descriptions in the revised manuscript.

Q14: All figures with LC traces need to mention the detection method in figures or legends. What are the signals shown in all figures?

A14: Many thanks for your kind reminder, and we have made related modifications in the revised manuscript.

Q15: PPM errors for all HRMS data need to be reported.

A15: Thanks for your kind reminder and PPM errors have been added in the Supplementary Methods 1.4 for HRMS data in the revised manuscript.

Q16: The scholarly presentation of manuscript needs to improve. There are numerous non-standard English and grammar and other mistakes. For example, line 253, Figure 2a. Line 260, Figure 2b.

A16: Many thanks for your reminders, and we have modified it in the revised

manuscript.

Q17: The TOC to start with AMP seems not appropriate. ATP should be used.

A17: Many thanks for your suggestion, and we have made corresponding modifications for the TOC in the revised manuscript.

Response to Reviewer 3

Q1: The AGM pathway proposed by the author is based on the results of one-pot enzymatic reactions in vitro. However, in a step-by-step in vitro experiment on AgmC-E-B, the starting substrate was Ribose-5-P, not Allulose-6-P in the AGM pathway. In addition, it is speculated that AgmD catalyzes Fructose-6-P to Allulose-6-P in the AGM pathway, which lacks direct evidence support (the substitution of AgmD with a homologous AlsE in the 6-enzyme one-pot reaction is not rigorous enough). Therefore, in order to solve the above problems, I think at least the following experiments need to be done:

1) One-pot reaction of AgmD and AgmC with Fructose-6-P as a starting compound, by detecting AMP production to determine whether Fructose-6-P can be used as a substrate of AgmD.

2) One-pot reaction of AgmD, AgmC, AgmE and AgmA with Fructose-6-P as a starting compound, by detecting compound 4 production to determine whether AgmE can catalyze the formation of the key glycosidic bond in AGMs.

A1: Many thanks for your valuable suggestions, and we have conducted the suggested experiments, and related data has been included in Figure S22.

Q2: Picture S10 is wrong. This picture is the same as Picture S9.

A2: Really sorry for this, and we have corrected Figure S10 in the revised manuscript.

Q3: In the manuscript or SI, there is no information about the detection wavelength

for HPLC analysis.

A3: Added as suggested.

Q4: Line 138, Please specify which gene cluster was cloned.

A4: Many thanks for your comments, and we have made corresponding modifications.

Q5: Line 143, M1154::pCHW501 repeated.

A5: Modified as suggested.

Q6: Line 178-180, The information of these two less critical proteins has been included in Table S1 and does not need to be described in the text.

A6: Thanks for your reminder, and we have made related modifications in the revised manuscript.

Q7: Line 211, dAMP dADP

A7: Modified as suggested.

Q8: Line 253, Figure 2a Figure 4a (?)

A8: Modified as suggested.

Q9: Line 259, Figure 2b Figure 4b (?)

A9: Modified as suggested.

Q10: Line 268, "Mycobacterium tuberculosis H37Rv (PDB: 3CE6)", is this SAH hydrolase a NAD⁺ self-sufficient enzyme like AgmF? The authors can discuss it from its catalytic mechanism and structural similarity.

A10: Thanks for your kind suggestion, and we have added related descriptions in the revised manuscript.

Q11: Line 305 and Figure S14, the phylogenetic analysis of AgmF and homologues does not include the structural model protein (Rv3248c from *Mycobacterium tuberculosis* H37Rv, PDB: 3CE6)? Which branch of the phylogenetic tree is Rv3248c located?

A11: Thanks for your reminders, and the structural model protein has already been added in the phylogenetic analysis of AgmF in the revised manuscript.

Q12: Line 326-330 and Figure S16d, the negative controls should include reactions without AgmC or AgmE.

A12: Many thanks for your suggestions, and we have added the related negative controls in the revised manuscript.

Q13: Figure S19, NADH+ NADP+

A13: Modified as suggested.

Reviewers' Comments:

Reviewer #1:

Remarks to the Author:

The authors have done well in addressing many of the concerns raised during the initial review of this manuscript. However, the mechanism shown in Figure 4E is still incorrect and does not make chemical sense. Either the authors should provide a more rational mechanism (one that does not involve water as a leaving group) or delete this panel from the figure. Once that task is completed, the manuscript should be acceptable.

Reviewer #2:

Remarks to the Author:

The revisions have reasonably addressed most of previous comments with some additional experiments and controls. The manuscript writing has improved in terms of clarity although some further proofreading can still be helpful.

Reviewer #3:

Remarks to the Author:

I am satisfied with this revision and support its publication.

Response to Reviewer 1

Q1: The authors have done well in addressing many of the concerns raised during the initial review of this manuscript. However, the mechanism shown in Figure 4E is still incorrect and does not make chemical sense. Either the authors should provide a more rational mechanism (one that does not involve water as a leaving group) or delete this panel from the figure. Once that task is completed, the manuscript should be acceptable.

A1: Many thanks for your kind suggestion, and we have made related modifications of Figure 4E in the revised manuscript (a detailed mechanism has been added as well to make the AgmF catalyzed dehydration step more understandable, **Supplementary Fig. 13C**). Moreover, we have also added the following discussions in the revised manuscript to make the proposed mechanism clearer to readers. "The leaving group proposed for the AgmF-catalyzed reaction is water, which is relatively poor when compared to the 5'-homocysteine motif for the SAH hydrolase family enzymes. However, this may be promoted by more favorable conditions within the active site of AgmF or, possibly involve a transient leaving group at the 5'-position."

Actually, dear reviewer, water as a leaving group in the dehydratase catalyzed reactions has been previously reported in several papers (ACS Catal, 2019 9(4):2962-2968; Biochemistry, 2017, 56(45):6030-6040.). Alternatively, it is also okay and acceptable for us to remove this panel (Fig. 4E) if you think something irrational is still existed in the updated mechanism for the AgmF catalyzed dehydration.

Response to Reviewer 2

Q1: The revisions have reasonable addressed most of previous comments with some additional experiments and controls. The manuscript writing has improved in terms of clarity although some further proofreading can still be helpful.

A1: Many thanks for your kind reminder, and we have carefully re-checked the manuscript to remove the potential typos.

Reviewers' Comments:

Reviewer #1:

Remarks to the Author:

The authors have now made the requested changes and corrections. Hence, this manuscript is suitable for publication.